# APPROXIMATE CLUSTERING FOR EXTRACTING TASK RELATIONSHIPS IN MULTI-INSTRUCTION TUNING

## ABSTRACT

The development of language models involves the evaluation of a broad range of learning tasks. Recent work has shown that by using carefully designed instructions to teach a large transformer model, they can be fine-tuned on a wide range of downstream tasks. However, when the number of instructions increases, they can negatively interfere with each other if trained together. Existing works have relied on domain expertise and manual inspection to construct multi-instruction sets, which can be time-consuming and difficult to scale. To address this challenge, this paper develops a clustering algorithm to find groups of similar tasks based on a given set of task affinity scores. This is an NP-hard problem, and conventional algorithms such as spectral and Llyod's clustering are sensitive to variations in the scale of task losses. Our algorithm instead uses a semidefinite relaxation to maximize the average density of clusters and then rounds the solution with a threshold. We adaptively build the clusters by gradually adding tasks so that the affinities only need to be computed in the existing clusters. Then, we construct an evaluation benchmark to assess task grouping algorithms with verified group structures. The evaluation set includes 63 cases, spanning multitask instruction tuning, multi-instruction tuning, and in-context learning of multiple functions. We validate our algorithm on this evaluation set by showing that it recovers the group structure found by an exhaustive search. We also show that our approach improves performance over multi-instruction and soft-prompt tuning by up to 6% on several sentence classification and structure-to-text generative tasks.

## 1 INTRODUCTION

A hallmark of the recent development in language models is that they can simultaneously make predictions over a broad range of learning tasks (Roberts et al., 2019; Liang et al., 2022). The adaptation of these language models to downstream tasks is then enhanced via instruction fine-tuning (Mishra et al., 2022). Prior work has shown that fine-tuning an existing model such as T5 through multiple instructions can lead to state-of-the-art results on a diverse collection of NLP tasks (Sanh et al., 2022; Wei et al., 2022). In light of these developments, the design of instruction tuning datasets and evaluations has received much interest recently (Longpre et al., 2023). By contrast, the algorithmic problem of how to best use these instructions for fine-tuning downstream tasks remains under-explored. It is also worth noting that these sets typically involve a large number of tasks and instructions, which can lead to severe negative interference when they are trained together naively (Jang et al., 2023). Existing work on multi-instruction tuning relies on extensive domain expertise and manual selection (Chung et al., 2022). In this paper, we revisit a task grouping problem (Standley et al., 2020), which is highly relevant to a number of settings in language model fine-tuning: Given $n$ tasks, we are interested in partitioning them into $k$ groups so that each group of tasks can be best trained together (separately from the other groups).

A naive approach to selecting which tasks to train together in a language model is according to the category of each task. Because the datasets are collected from different sources (Wang et al., 2018; Aribandi et al., 2022), even two tasks of the same category, such as sentiment analysis, may not transfer positively to each other. Task grouping methods have been developed for jointly learning multiple datasets. For instance, Fifty et al. (2021) first compute a set of (pairwise) affinity measures and then apply optimization techniques such as branch-and-bound to find the best task combinations. The computational cost of these techniques can still be quite high for the scale of instruction fine-

tuning sets. Another natural solution is to use clustering algorithms such as spectral clustering (Ng et al., 2001) and Lloyd's algorithm (Lloyd, 1982). We find that these methods are particularly sensitive to the scale of the varied losses across a large set of different tasks.

To address the challenge, we develop a new clustering algorithm, which involves two key steps. The first step is a semidefinite relaxation for maximizing the average density of the $k$ groups, given an $n$ by $n$ task affinity matrix $T$. This matrix requires measuring $n^2$ affinity scores, which can be slow to compute when $n$ is large. Therefore, the second step of our algorithm is an adaptive procedure, where we build the clusters gradually. This allows us to accelerate the computation of task affinities by leveraging the existing separations in the clusters. Moreover, we introduce an adaptive sampling technique to account for higher-order task relationships.

To facilitate the evaluation of task grouping methods, we curate an evaluation benchmark that contains task group structures with verified positive transfer within groups. This benchmark includes 63 evaluation cases that span three types of scenarios, including multitask (instruction) fine-tuning (over 19 NLP tasks) (Wang et al., 2018, 2019; Sanh et al., 2022), multi-instruction fine-tuning (over 100 instructions) (Bach et al., 2022; Zhou et al., 2023), and in-context learning with three function classes (Garg et al., 2022). See Tab. 1 for a summary. Based on this benchmark, we evaluate our approach by showing that the above algorithm can correctly identify the underlying groups, succeeding in all evaluation cases. Notably, the groups match the results found by exhaustive search. We also show that our approach outperforms multi-instruction and prefix tuning by **3.3%** on three sentence classification tasks from SuperGLUE (Wang et al., 2019) and two structure-to-text generative tasks from the GEM benchmark (Gehrmann et al., 2021).

In summary, in this paper, we revisit the task grouping problem for language model fine-tuning and design a new clustering algorithm that is both efficient and robust to cross-task heterogeneity. We construct an evaluation benchmark for task grouping approaches along with an easy-to-use package, spanning three scenarios of instruction fine-tuning, which can also be used for future work. Experiments show that our algorithm can correctly identify the underlying group structures and also be used to identify groups of similar instructions in multi-instruction tuning. The rest of this paper is organized as follows. Sec. 2 reviews related work. Sec. 3 provides more background information. Sec. 4 describes our algorithm. Sec. 5 presents the experiments. Additional related work and experiment details are provided in the Appendix.

## 2 RELATED WORK

Previous works of FLAN (Wei et al., 2022), NaturalInstructions (Mishra et al., 2022), and T0 (Sanh et al., 2022) have demonstrated that fine-tuning language models on multiple downstream tasks prompted with instructions resulting in enhanced generalization to previously unseen tasks. Moreover, there have been efforts to advance the instruction fine-tuning method, such as expanding task datasets (Chung et al., 2022; Wang et al., 2022b) and refining instruction sets (Bach et al., 2022). Furthermore, Muennighoff et al. (2023) constructed multilingual instruction fine-tuning datasets and performed instruction fine-tuning on multilingual pretrained language models to boost the generalization on unseen language and tasks. Longpre et al. (2023) study the design decisions of publicly available instruction tuning methods and find that training with mixed instructions settings yields improved performance. Wang et al. (2023b) propose multitask prompt tuning, which first learns a single transferable prompt by distilling knowledge from multiple task-specific source prompts and then learns multiplicative low-rank updates to this shared prompt to adapt it to each downstream target task efficiently. Compared to these studies, we study the algorithmic problem of how to find task structures in instruction fine-tuning sets.

There is also a growing line of work on designing the best prompt to adapt pre-trained language models to downstream tasks (Shin et al., 2020; Gao et al., 2021; Zhang et al., 2022). Prefix tuning (Li and Liang, 2021) inserts continuous prompt embeddings to each layer in language models and optimizes the embeddings during fine-tuning. Prompt tuning (Lester et al., 2021) proposes to add prompt embeddings only in the inputs. PromptBoosting (Hou et al., 2023) constructs a large pool of weak learners by pairing prompts with different elements of the LM's output distribution and then ensemble the weak learners using the AdaBoost algorithm. Instead of finding the best instruction for a downstream task, our work focuses on optimizing the average performance of a model under multiple instructions.

Table 1: We construct 63 evaluation cases with verified group structures to assess task grouping algorithms. Below is a table of summary of the category of tasks and datasets included in each evaluation set.

| | | | |
|---|---|---|---|
| Multitask fine-tuning | 57 evaluation cases with 2-6 (varied) groups | | |
| | Sentiment Classification (3)   Natural Language Inference (3)   Multiple-Choice QA (4) | | |
| | Open-Domain QA (3)    Coreference Resolution (3)    Summarization (3) | | |
| Multi-instruction fine-tuning | 5 evaluation cases with 10 groups in each | | |
| | RTE (100)   WiC (100)   BoolQ (100)   E2E NLG (100)   Web NLG (100) | | |
| In-context learning | 1 evaluation case with 3 groups | | |
| | Linear Regression (3)   Decision Trees (3)   Neural Networks (3) | | |

Clustering is a fundamental aspect of machine learning. Besides semidefinite programming relaxations, linear programming relaxations are known for clustering objectives such as $k$-center (Guttmann-Beck and Hassin, 2000). Their approach requires pre-selecting $k$ anchor points as the centers of each cluster. However, their approach then enumerates all $k$-sized subsets and thus runs in time $O(n^k)$. For the case of geometric clustering in Euclidean spaces, polynomial time approximation schemes can be achieved (De La Vega et al., 2003). Bartal et al. (2001) give a polynomial time approximation algorithm for min-sum $k$-clustering for arbitrary metric spaces via dynamic programming. The integrality gap of linear programming and semidefinite programming relaxations can be analyzed when there is a separation structure in the underlying clusters (Awasthi et al., 2015). These approximation guarantees typically require the underlying similarity scores to satisfy a metric condition. By contrast, the task affinity matrix, in our case, can easily violate the triangle inequality. Lastly, recent work has also looked into mixed integer programming for best subset selection (Bertsimas et al., 2016). One novel contribution of this work is to make explicit a connection between multitask/multi-instruction fine-tuning and clustering. In light of this connection, it would also be interesting to revisit hierarchical clustering (Charikar and Chatziafratis, 2017; Chami et al., 2020) and hypergraph clustering (Yin et al., 2017; Veldt, 2023) for task grouping.

## 3   PRELIMINARIES

Many problems in the context of language model fine-tuning are related to multitask learning (Wang et al., 2018; Aribandi et al., 2022; Sanh et al., 2022). We give three examples, which will be the focus of this paper: (1) Multitask instruction fine-tuning is an essential component of adapting language models, enabling the models with various language processing abilities, such as question answering and text summarization. (2) Multi-instruction fine-tuning involves a mix of instructions to further enhance a language model's ability to respond to diverse instructions from users. (3) In-context learning refers to the ability of a language model to learn a function class with a few "in-context" examples; A natural question is whether these different function classes are in-context learnable simultaneously.

**Task Grouping Setup.** The above examples can be formulated abstractly in a multitask learning setting. Let there be $n$ downstream tasks. The goal of task grouping (cf. Standley et al. (2020)) is to partition the $n$ tasks into $k$ subsets such that each subset of tasks is the best to be trained together.

For each pair of tasks $u$ and $v$, let $T_{u,v}$ denote an affinity score, which quantifies the transfer effect between them. Pairwise notions of affinity scores between two tasks have been used in prior work (Fifty et al., 2021). For example, one way to quantify $T_{u,v}$ is via the performance of task $u$'s validation performance evaluated on a model fine-tuned on a model trained with both $u$ and $v$. Given an $n$ by $n$ task affinity matrix $T$, the extent of positive transfers within a subset of tasks $S$ can be characterized by the density of affinity scores in the subset:

$$d_S = \sum_{u,v \in S} \frac{T_{u,v}}{|S|}. \tag{1}$$

Then, one can view task grouping as a clustering problem whose objective is to maximize the average density of all clusters. Let $C_1, \ldots, C_k$ denote a partition of the $n$ tasks. Let $v_1, \ldots, v_k$ be a 0-1

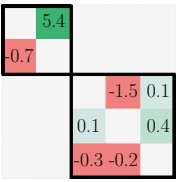 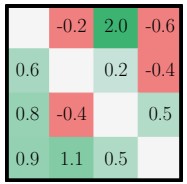 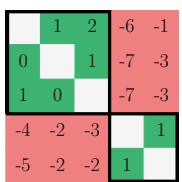

(a) Pairwise transfers for two groups of tasks from GLUE

(b) Pairwise transfers between four NLI tasks from GLUE

(c) Pairwise transfer results for five different instructions

Figure 1: We illustrate negative interference between tasks. For each entry, we pick one task as the target task, combine it with another task, and report the performance difference between multitask and single-task learning. We also notice negative interference between instructions. Fine-tuning with two instructions may decrease the performance of a single instruction.

vector indicating whether each task is in the cluster or not. The average density can be written as:

$$\frac{1}{k}\sum_{i=1}^{k}d_{C_i} = \frac{1}{k}\sum_{i=1}^{k}\sum_{u,v\in C_i}\frac{T_{u,v}}{|C_i|} = \frac{1}{k}\sum_{i=1}^{k}\frac{v_i^\top T v_i}{v_i^\top v_i}. \tag{2}$$

This is an integer program, which is NP-hard to optimize in general (in particular, it contains the geometric clustering problem as a special case (Aloise et al., 2009)). Previous work (Fifty et al., 2021) has proposed branch-and-bound methods to solve this, which is still computationally expensive.

**Negative Interference.** We verify the existence of negative interference in the examples. First, we fine-tune a language model on nine NLP tasks in the GLUE benchmark (Wang et al., 2018), which classifies them into three groups, including two single-sentence tasks (CoLA and SST-2), three similarity and paraphrase tasks (MRPC, QQP, and STS-B), and four NLI tasks (MNLI, QNLI, RTE, and WNLI). We examine the pairwise transfers by fixing one task as the target and the rest as the source. We fine-tune a RoBERTa-Base model, combining one source task with the target. We evaluate the performance difference between multitask and single-task models on the target task's dev set. Second, we fine-tune a language model with multiple instructions. We view one instruction as one task. We compute pairwise transfers between instructions. We use five instructions from PromptSource (Bach et al., 2022) and fine-tune a T5-Base model on the RTE dataset from SuperGLUE. Each time, we fine-tune a model with two instructions and compare its performance with the model fine-tuned with a single instruction. In Fig. 1, each row corresponds to one target task. The entries below zero correspond to negative transfers. We observe a mix of positive and negative transfers, motivating the need to develop evaluation sets for task grouping.

## 4 ALGORITHM

We now describe our algorithm for maximizing the average density of the task group. We develop a semidefinite programming (SDP) relaxation and then generate clusters by rounding the SDP solution above a threshold. Then, we design an adaptive grouping procedure that builds clusters gradually.

### 4.1 SEMIDEFINITE PROGRAMMING RELAXATIONS FOR TASK AFFINITY CLUSTERING

To maximize the objective stated in Eq. (2), we can use an assignment variable from every task to every cluster. More precisely, let us denote the assignment variables as an $n \times k$ matrix $V$, such that each entry $V_{i,j}$ indicates whether a task $i$ belongs to a cluster $j$, for every $i = 1,\dots,n, j = 1,\dots,k$. Moreover, let the $j$th column of $V$, which is the characteristic vector of the $j$-th cluster, be denoted as $v_j$. Under this assignment, the sum of $V_{i,j}$ across any task $i$ must be one, as we allow one task to be assigned in a single group. By contrast, the sum of $V_{i,j}$ across a cluster $j$ is the number of tasks assigned to the $j$-th cluster, which will be at least one.

Next, we state an integer program to maximize the average density of all $k$ clusters in Eq. (2):

$$\max\left\{\langle T, \sum_{j=1}^{k}\frac{v_j v_j^\top}{v_j^\top v_j}\rangle : Ve = e, \sum_{i=1}^{n}V_{i,j} \geq 1 \text{ for } 1 \leq j \leq k, V \in [0,1]^{n\times k}\right\}, \tag{3}$$

where $e$ is the all-ones vector. We omit the $\frac{1}{k}$ factor in the objective for simplicity.

This integer program is computationally challenging to solve, even for small values of $k$. To address this issue, we will relax the above integer program to a (constrained) semidefinite program (SDP), which can be solved in polynomial time. First, we note that $v_i v_i^\top$ is a rank one semidefinite variable. Let us denote the sum of them (normalized by $v_i^\top v_i$) as the following new variable

$$X = \sum_{j=1}^{k} \frac{v_j v_j^\top}{v_j^\top v_j}. \tag{4}$$

This matrix $X$ has a rank equal to $k$ because it is the sum of $k$ rank-1 matrices, and the $v_i$'s are orthogonal to each other. Additionally, its trace is equal to $k$ because $\frac{v_j v_j^\top}{v_j^\top v_j}$ has a trace of one for any $j$. Second, the entries of every row of $X$ is equal to one:

$$Xe = \sum_{i=1}^{k} \frac{v_i (v_i^\top e)}{v_i^\top v_j} = \sum_{i=1}^{k} v_i = e.$$

Removing the 0-1 integer constraint, we relax Problem (3) into a rank-constrained problem:

$$\max \left\{ \langle T, X \rangle : Xe = e, \operatorname{rank}(X) = k, \operatorname{Tr}[X] = k, X \geq 0, X \succeq 0, X \in \mathbb{R}^{n \times n} \right\}.$$

The above program involves a rank constraint, which is still computationally challenging to solve. However, it can be further relaxed by removing the rank constraint while keeping the trace constraint:

$$\max \left\{ \langle T, X \rangle : Xe = e, \operatorname{Tr}[X] = k, X \geq 0, X \succeq 0, X \in \mathbb{R}^{n \times n} \right\}. \tag{5}$$

The above problem can be solved efficiently using convex optimization methods. Given a solution of $X$, the last step is to round it into an integer solution. We set a threshold $\lambda$ such that if $X_{u,v} \geq \lambda$, tasks $u$ and $v$ are assigned to the same cluster. In practice, we set the $\lambda$ as $c/n$ for some constant $c \geq 1$, since $X_{u,v}$ should be $\frac{1}{|C_i|}$ when they are in the same cluster $C_i$. In summary, we can derive an efficient clustering algorithm given a task affinity matrix; See Procedure 1 below.

---

**Procedure 1** Approximate Task Clustering though SDP Relaxations

---

**Input**: Task affinity matrix $T \in \mathbb{R}^{n \times n}$
**Require:** Number of clusters $k$; A threshold $\lambda$ for rounding
**Output**: A list of clusters $\mathcal{C}$
 1: Obtain $X$ by solving problem (5)
 2: Generate a list of clusters $\mathcal{C}$ by assigning $u$ and $v$ into a cluster if $X_{u,v} > \lambda$

---

**Illustrative Example.** A naive way to maximize the clustering objective is using algorithms such as spectral clustering or Lloyd's algorithm on the task affinity matrix $T$. Curiously, we observe that these algorithms are not robust in multitask learning as the scale of different tasks' losses varies dramatically. In Fig. 2, we illustrate the clustering results with these methods. We use a planted model by generating a random matrix including one low-density cluster and two high-density clusters.

- In spectral clustering, the eigenvector values remain constant on the high-density clusters with the presence of the low-density cluster.
- Lloyd's algorithm iteratively selects the cluster centroids and updates assignments to each cluster. With higher values in high-density clusters, the centroids are assigned to them, and the algorithm does not separate the low-density cluster.

## 4.2 Adaptively Estimating Task Affinities and Building Task Clusters

Next, we design an algorithm to speed up the clustering process. The above clustering algorithm requires access to the pairwise task affinity matrix. For $n$ tasks, computing the pairwise affinity scores between every pair of tasks is time-consuming, as it requires training $n^2$ models. Furthermore, it ignores higher-order task relationships beyond the combinations of two tasks, which adds more

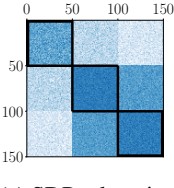 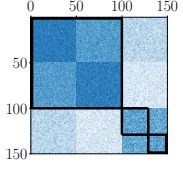 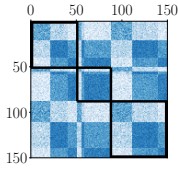

(a) SDP relaxation       (b) Spectral clustering       (c) Lloyd's clustering

Figure 2: We illustrate the SDP relaxation compared to spectral clustering and Lloyd's algorithm for recovering three hidden clusters. Spectral clustering groups the two higher-density clusters together, while Lloyd's algorithm mixes the three clusters. By contrast, the SDP relaxation manages to identify all three hidden clusters. The black solid line illustrates the clusters yielded by each algorithm. In this example, we generate three clusters with three levels of densities, each with 50 data points.

complexity to multitask relationships. We first introduce a task affinity score that captures higher-order task relationships over task subsets. Then, we design an efficient sampling procedure by iteratively computing part of the affinity matrix and growing the clusters adaptively.

**Higher-Order Task Affinity.** We consider a higher-order task affinity score estimated from subsets of more than two tasks. First, sample $m$ subsets of tasks $\{1, 2, \ldots, n\}$ uniformly over subsets of size $\alpha$, denoted as $S_1, S_2, \ldots, S_m$. Then, compute the multitask learning performance (e.g., accuracy) of task $i$, denoted as $f_i(S_j)$, by fine-tuning a model on tasks in every subset $S_j$ for $j = 1, \ldots, m$. Lastly, compute $T_{i,j}$ as the average multitask performance over all subsets that include task $i$ and $j$:

$$T_{i,j} = \frac{1}{n_{i,j}} \sum_{1 \leq k \leq n : \{i,j\} \subseteq S} f_i(S_k), \text{ for all } 1 \leq i, j \leq n, \tag{6}$$

where $n_{i,j}$ be the number of subsets that include both $i, j$. This sampling is analogous to the sampling of features in random forests (due to space limit, a detailed justification is stated in App. B.3).

**Adaptive Sampling.** The next step is an adaptive sampling procedure to accelerate the above estimation. The idea is to divide tasks into small batches and iteratively estimate affinity scores for a new batch of tasks. In each iteration, we have existing cluster structures and a new batch of unclustered tasks. We pick one cluster to estimate task affinity scores between the chosen cluster and the new batch of tasks. This uses the existing separations, as described in Procedure 2. After estimat-

---

**Procedure 2** Adaptive Estimation of Task Affinity Scores

**Input**: $n$ tasks, training and validation sets of each task, cluster structure $\mathcal{C}_0$ for the first $n_0$ tasks
**Require:** Number of subsets $m$; Size of each subset $\alpha$; Multitask learning algorithm $f$
**Output**: An $n$ by $n$ task affinity matrix $T$

1: **for** $i = 1, 2, \ldots, m$ **do**
2:      Randomly choose a group $C$ from cluster $\mathcal{C}_0$
3:      Sample a random subset $S_i$ from $\{n_0 + 1, n_0 + 2, \ldots, n\} \cup C$ with size $\alpha$
4:      Evaluate multitask performance $f(S_i)$ for every task in $S_i$
5: **end for**
6: Calculate the affinity score matrix via Eq. (6)

---

ing the affinity scores for the new batch of tasks, we update the clusters by solving the relaxed SDP in Eq. (5). We initialize the task assignment variable $X$ by assigning $X_{u,v}$ as $\frac{1}{|C|}$ if $u$ and $v$ are in a cluster $C$ with size $|C|$. Then, we solve the SDP again to re-generate the clusters. At iteration $t$, we search the number of clusters within a range of $|\mathcal{C}^{(t)}|$ to $k$ and choose the one that maximizes the objective $\langle T, X \rangle$. The complete procedure is described in Algorithm 3.

**Runtime.** We examine the runtime of our algorithm. There are $s$ iterations. During each iteration:

- We estimate task affinity scores for $b$ tasks. We train $m$ models on sampled subsets to compute the scores. In practice, we notice that collecting $m = 5b = \frac{5n}{s}$ subsets suffices for estimating the affinity scores until convergence. For $n = 100$ tasks, we take $s = 10$ steps. Each step trains 50 models on sampled subsets and takes 23 hours using a single GPU.

- We solve a convex program on an affinity matrix of size $n$ by $n$. In practice, this step typically runs quickly in our experiments, taking less than 1.5 seconds for $n$ up to 100.

---

**Algorithm 3 Adaptive Task Grouping (AdaGroup)**

---

**Input**: $n$ tasks, training and validation datasets of each task
**Require:** # final clusters $k$; # adaptive steps $s$; # sampled subsets in each step $m$; Size of subset $\alpha$
**Output**: $k$ groups of tasks

 1: Initialize the clusters as $\mathcal{C}^{(0)} = \{\}$. Let $b = \frac{n}{s}$ be the number of additional tasks in each step
 2: **for** $t = 0, 1, \ldots, s-1$ **do**
 3:     Choose $b$ tasks from the remaining tasks
 4:     Estimate the task affinity matrix $T^{(t+1)}$ by Procedure (2) with current cluster structure $\mathcal{C}^{(t)}$
 5:     Generate clusters $\mathcal{C}^{(t+1)}$ following Procedure (1)
 6: **end for**
 7: **return** $\mathcal{C}^{(s)}$

---

## 5 EXPERIMENTS

We describe experiments to apply our algorithm to three problems relevant to language model fine-tuning, including multitask fine-tuning, multi-instruction tuning, and in-context learning. We will discuss the evaluation datasets used in the experiments. Then, we describe the setup along with the comparative results. Lastly, we give ablation studies to justify our algorithm design and end this section with a discussion for future work.

### 5.1 EVALUATION OF TASK GROUPING

The evaluation of task grouping algorithms requires a clear specification of task grouping structures. A naive way to conduct evaluations is using existing multitask learning benchmarks such as GLUE (Wang et al., 2018) and SuperGLUE (Wang et al., 2019). These benchmarks come with pre-defined groups. Curiously, we noticed that nearly 40% of pairwise transfers are negative even within these groups, as shown in Fig. 1. With this context in mind, the first aim of our experiments is to collect and then construct an evaluation benchmark that is more suitable for assessing task grouping algorithms. Ideally, such an evaluation set should have clearly defined group structures.

**Multitask Instruction Fine-Tuning**. We collect a list of NLP datasets under different (human-labeled) categories, such as sentiment analysis, question answering, summarization, etc. Then, we measure the pairwise transfers between each pair of tasks from the same category. We use T5-Base as the base model (Raffel et al., 2023).

After getting all the pairwise effects for each category, we select the subsets whose ratio of positive effects is higher than 90%. This leads to an evaluation set of six groups of six task categories. These include sentiment analysis, natural language inference, multiple-choice QA, open-domain QA, coreference solution, and summarization tasks. Each category contains three or four tasks, leading to 19 tasks in total. We display a complete list in Table 3 (App. B).

**Multi-Instruction Tuning**. We consider three datasets from SuperGLUE, including RTE, WiC, and BoolQ, and two structure-to-text generation datasets from the GEM benchmark (Gehrmann et al., 2021), including E2E NLG challenge and Web NLG. Each dataset contains 100 instructions, including ten instructions from Bach et al. (2022) and 90 instructions that we generate with an automatic instruction generation method from Zhou et al. (2023).

**In-Context Learning**. We define one in-context learning task for one function class, following the setup in Garg et al. (2022). Each task contains sequences of $d$ in-context examples, denoted as $(x_1, \phi(x_1), x_2, \phi(x_2), \ldots, x_d, \phi(x_d))$ where $\phi$ is a random function sampled from the function class. We consider three types of functions, including linear regression (LR), decision trees (DT), and two-layer ReLU neural networks (NN). For each type, we define three function classes with different distributions. For example, for each function class of linear regression, we specify a Gaussian distribution over the weight parameters. In total, there are nine tasks corresponding to three groups.

### 5.2 IMPLEMENTATION AND BASELINES

For multitask instruction fine-tuning, we create evaluation cases and verify the group structure inside each case. Altogether, we have 15 cases with two groups, 20 cases with three groups, 15 cases with four groups, 6 cases with five groups, and 1 case with six groups. To verify that the group structure is

Table 2: Accuracy and Rouge1 scores on the development set averaged over all instructions on three sentence classification tasks from SuperGLUE and two structure-to-text generative tasks from GEM. We compare our approach with multi-instruction tuning, prefix tuning, and prompt tuning. we report the average results over three random seeds.

| Dataset | RTE | WiC | BoolQ | E2E NLG | Web NLG |
|---|---|---|---|---|---|
| Task Type (Metric) | Classification tasks (Accuracy) | | | Generative tasks (ROGUE-1) | |
| Multi-Instruction Tuning | 75.09±0.68 | 66.44±0.98 | 78.16±0.77 | 71.46±0.27 | 80.80±0.19 |
| Prefix Tuning | 72.74±2.40 | 62.29±2.93 | 76.19±0.98 | 70.23±0.40 | 78.69±0.26 |
| Prompt Tuning | 73.12±1.26 | 62.88±2.19 | 75.51±0.85 | 70.72±0.81 | 77.42±0.31 |
| Our Approach | **80.96±0.85** | **69.89±0.87** | **81.76±0.62** | **73.03±0.67** | **82.95±0.75** |

correct, we use an exhaustive search to enumerate all task combinations that optimize the clustering objective (cf. Eq. (3)) and make sure that the group structure indeed achieves the optimum for the clustering objective.

For multi-instruction fine-tuning, we use T5-Base as the base model. For classification tasks, we report the accuracy as the performance. For generative tasks, we report the Rouge1 score as the performance. For each dataset, we evaluate the average performance over all 100 instructions. In our approach, we view one instruction as one task. We apply our approach to find groups of instructions and then fine-tune one model for each group of instructions. Our approach requires three hyperparameters: the number of adaptive steps, the number of subsets in each step, and the size of subsets. We select the size between 3, 5, and 10. We select adaptive steps between 10, 5, and 3. We then set the number of subsets as five times the number of new tasks in each step. We select the number of clusters from a range between 10, 15, and 20.

We compare our approach with multi-instruction and report the results of two soft-instruction tuning baselines in terms of relative improvement, including Prefix Tuning (Li and Liang, 2021) and Prompt Tuning (Lester et al., 2021). We use LoRA fine-tuning (Hu et al., 2022) for our approach and multi-instruction to match the same amount of training parameters as soft-instruction tuning. The information for the training details is included in Appendix B.

For in-context learning, a transformer is trained to predict $\phi(x_i)$ for a given $x_i$ based on the preceding in-context examples. For each task, we evaluate the prediction loss as the squared error of predictions averaged over $d = 100$ in-context learning steps and use this loss as the MTL performance $f(S)$. For estimating task affinity scores between tasks, we sample subsets of three tasks, train a transformer on the examples from the three tasks, and evaluate the prediction loss on each task.

## 5.3 EXPERIMENTAL RESULTS

**Multitask Instruction Fine-Tuning Results.** We evaluate our approach on the 57 evaluation cases ranging from two to six groups of tasks. Our approach correctly identifies the underlying groups under all cases, obtaining the same results as the exhaustive search. In contrast, using spectral and Lloyd's clustering correctly identifies the group structures in 16/4 out of the 57 cases.

**Multi-Instruction Tuning Results.** Table 2 shows the results of the average performance on the development set evaluated with 100 instructions. We observe that our approach improves over the baseline methods by 3.3% on average, suggesting the benefit of separating instructions to reduce their negative interference.

**In-Context Learning Results.** We observe that transformers trained on different function classes perform worse than being trained on a single function class. We illustrate the task affinity scores between the function classes in Figure 3 (Left). The functions of the same type have larger task affinity scores than functions of different types. Our approach recovers the cluster structure for three types of function classes. In contrast, spectral clustering and Llody's clustering yield clusters mixed between different function classes, shown in Figure 3 (Right).

Figure 3: Clusters of function classes generated by our approach (Left), spectral clustering (Middle), and Lloyd's clustering (Right). Each entry corresponds to an affinity score using the mean squared loss as the MTL performance (green means a positive transfer, while red means a negative transfer).

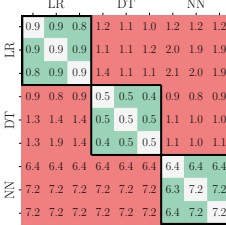 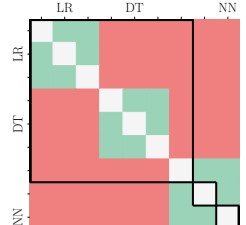 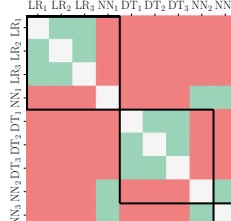

## 5.4 ABLATION STUDIES

We provide two ablation studies of our algorithm, including the clustering step and task affinity. Then, we illustrate an intriguing transfer between function classes during in-context learning.

**Instruction Selection.** We compare our clustering algorithm with alternative clustering methods, including spectral clustering and Lloyd's clustering. We find that our algorithm outperforms the two conventional clustering methods by 5.2%, averaged over the datasets in multi-instruction tuning.

**Task Affinity.** We compare alternative methods to estimate task affinity scores and validate the benefit of using higher-order task affinity. We compare the higher-order task affinity with two pairwise task affinity scores, including loss-based pairwise affinity (Standley et al., 2020), and gradient-based affinity score (as the ratio of task $i$'s loss before and after applying the gradient of task $j$ on the model parameters) (Fifty et al., 2021). We find that using higher-order task affinity improves the performance of grouping instructions by 1.7% over the two pairwise affinity scores on average.

**In-Context Transferability.** We examine the prediction loss of one type of function when training a transformer with another type of function. We first train a transformer only on examples of neural network functions (STL). Then, train a transformer on the combined training examples with another function class, including linear regression or decision trees (MTL). We compare the error between MTL and STL evaluated on examples of neural network functions in Fig. 4. Curiously, we find that a transformer trained with linear regression or decision trees compares comparably to a transformer only trained on neural networks. On the other hand, if we evaluate the performance on decision tree or linear regression, training a transformer with neural networks will significantly degrade MSE (see Fig. 6 of App. B.2).

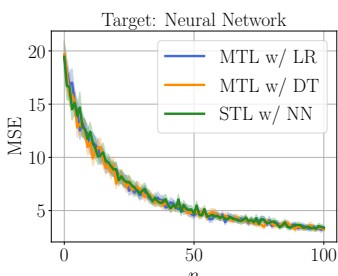

Figure 4: Test MSE of in-context learning neural network functions.

## 5.5 DISCUSSIONS

Our findings provide some evidence that modeling task relationships can also enhance language modeling (particularly instruction tuning). It might be interesting to investigate if this perspective applies in other contexts, such as modeling the relationship between generating different programming languages or algorithmic reasoning. It may also be worth investigating hierarchical relationships: Expriments shows that learning NNs implies the learning of linear regression and decision trees. It is plausible to revisit curriculum learning for tuning instructions with increasing complexities. To facilitate the discussion, we provide an easy-to-use package to make our evaluation sets accessible to researchers.

## 6 CONCLUSION

This paper developed an approximate clustering algorithm to extract task group structures so that the most related tasks are trained together. We construct a new evaluation benchmark for this clustering problem, spanning three use cases of language model fine-tuning, with a total of 63 evaluation cases. A package is developed for reusing this evaluation set to facilitate future discussions.

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
