# OpenReview forum: "Approximate Clustering for Extracting Task Relationships in Multi-Instruction Tuning"
_ICLR.cc/2024/Conference — ICLR 2024 Conference Withdrawn Submission_

### Official Review · Reviewer_oSbm · 2023-10-31

**Soundness:** 3 good
**Presentation:** 2 fair
**Contribution:** 3 good
**Rating:** 6
**Confidence:** 3

**Summary:**

The authors are studying the problem of identifying which tasks should be mixed together during training such that there is a positive transfer between these tasks and they help each other to improve. The authors first assume access to a pairwise task affinity matrix which specifies the improvement on task j when task i is trained before it. Assuming access to this task-affinity matrix they formulate the clustering problem in terms of maximizing the average density of task affinity scores across all clusters which is NP-hard. Then the authors use SDP relaxations to arrive at an approximate task clustering algorithm AdaGroup. They discuss and compare spectral and Lloyd clusters as two other approaches.

**Strengths:**

S1. The problem of task clusters is extremely important and hard to work with because the ground truth clusters are mostly not available. Grouping tasks appropriately can lead to significant improvements in the final model's performance.

S2. The method is well-motivated and can adaptively add new tasks to the existing cluster.

S3. The authors have computed the pairwise affinity of some NLP tasks and released them as a benchmark so that the community can try out different ideas to cluster tasks.

**Weaknesses:**

W1 The paper is slightly hard to understand, especially the experimental section. Many statements about

W2. New tasks can be processed in a batched manner to avoid expensive computations, however, the number of clusters are usually predefined and new tasks cannot allocate new clusters. This seems like a problem to me, some non-parametric Bayesian processes like the Chinese restaurant process, and Indian buffet process can potentially be used to solve this issue. Defining the number of clusters can be a challenging as well as computationally expensive process as multiple models are needed to be trained to obtain the clusters for a new set of tasks.

W3. The paper seems to talk about spectral and LLoyd clusters but in Table 2, I don't see them as a baseline. For example, you could use the clustering obtained from spectral and Lloyd algorithm and then perform training similar to your method. This I feel is an important baseline as it would tell us the impact of identifying wrong task clusters. It might also be the case that the impact of having wrong clusters is not too high, i.e. some decent amount of noise in the clusters does not impact the final downstream performance.

**Questions:**

**Must do for me to retain my scores**

Q1: Please improve the writing of the in-context learning experiments, I am not able to understand the motivation of, the experimental setup, and the conclusions for it. I am still not sure if I understand where logistic regression and decision trees come into this picture and the implications of the designed experiments.

Q2: same as W3. W3 needs to be addressed for retaining the score.


**Answer these for me to consider increasing my scores**

Q3: On the created benchmark AdaGroup can identify all the clusters correctly. It would be nice to see if you take some heldout tasks, and then use your AdaGroup to cluster these tasks and visualize the cluster/affinity scores. It would be nice to see how much these clusters correlate with human understanding. If this correlation is high then for a small number of tasks doing the clustering manually might be a reasonably good choice.

Q4: Given that the method needs to train multiple models in order to estimate the task affinity matrix, this method might not be feasible in cases where the datasets are pretty huge. It would be a really nice study to see what is the minimum number of samples from each task that you can use in order to get reliable task affinity scores that can lead to good clusters and improved downstream performance. This would ameliorate the costs associated with this method and make it scale to more number of tasks.

Q5: A solution to the cluster number number problem, see W2 for more details.

---

> ### Author Response · Authors · 2023-11-18
> **Response to Reviewer oSbm 1/2**
>
> We thank the reviewer for the detailed feedback. We respond to the questions raised by the reviewer in detail below.
>
> \>>> **Clarifying the in-context learning experiments.**
>
> We apologize for the confusion in our initial submission. We include in-context learning as a test case in the task grouping evaluations. The design of the in-context learning (ICL) task follows the recent paper by Garg et al. (2022):
>
> - We view a task as in-context learning for one function class. A task contains in-context example sequences $(x_1, \phi(x_1), x_2, \phi(x_2), \ldots, x_d, \phi(x_d))$ generated by a function $\phi$ sampled from a function distribution.
> - We consider three types of functions, thus naturally forming three groups of tasks, which are linear regression, random forest, and neural networks. We define three tasks within each group, corresponding to three function distributions. In total, there are nine tasks.
>
> Our hypothesis is that training a model on tasks of different function types will result in negative interference. Therefore, ICL tasks of the same function type have larger task affinity scores than different function types. One clustering algorithm passes this case when functions of the same type are grouped into the same cluster.
>
> Our results first confirm that joint training ICL tasks of the same function class result in lower loss than training on two different types of functions. This results in larger task affinity within groups than inter-groups. Second, we find that based on such an affinity score matrix, our algorithm can accurately identify the three types of functions, while the previous two clustering algorithms cannot. We will revise the experiments to make this clearer in the updated paper.
>
> \>>> **Adding the results of spectral and Lloyd’s clustering.**
>
> Thanks for the suggestion. We have added the results of spectral and LLoyd’s algorithms for grouping instructions in multi-instruction fine-tuning. We set the number of groups the same as in our approach. Indeed, we find that our approach outperforms these two cluster algorithms by 5.3% on average over the five datasets. This aligns with our observation that spectral and LLoyd’s algorithm is sensitive to scale variations among clusters and can lead to wrong clusters.
>
> | Dataset                             | RTE                | WiC                | BoolQ              | E2E NLG            | Web NLG            |
> | ----------------------------------- | ------------------ | ------------------ | ------------------ | ------------------ | ------------------ |
> | Metric                              | Accuracy           | Accuracy           | Accuracy           | ROGUE-1            | ROGUE-1            |
> | Grouping by Spectral Clustering     | 73.18$\pm$0.45     | 65.09$\pm$0.67     | 75.71$\pm$0.55     | 71.91$\pm$0.26     | 81.27$\pm$0.94     |
> | Grouping by Lloyd's Clustering      | 73.58$\pm$1.03     | 64.61$\pm$0.64     | 75.61$\pm$0.30     | 71.26$\pm$0.42     | 80.41$\pm$0.49     |
> | Clustering by SDP relaxation (Ours) | **80.96$\pm$0.85** | **69.89$\pm$0.87** | **81.76$\pm$0.62** | **73.03$\pm$0.67** | **82.95$\pm$0.75** |
>
> We will incorporate the results into the updated paper.

---

> ### Author Response · Authors · 2023-11-18
> **Response to Reviewer oSbm 2/2**
>
> \>>> **Determining the number of clusters… Some non-parametric Bayesian processes like the Chinese restaurant process and Indian buffet process can potentially be used to solve this issue…”**
>
> We thank the reviewer for the comment. In our approach, after estimating task affinities for new tasks, we use a grid search approach to determine the number of clusters. We choose the number of clusters with the highest average density of clusters, which is the objective of our clustering algorithm. We notice that the average density of clusters correlates with the resulting performance, thus making it a useful criterion.
>
> To illustrate this, we ablate the number of clusters for clustering the 100 instructions on the RTE dataset. We vary the number of clusters between 5, 10, 15, 20. Then, we evaluate the average density of clusters and the test performance of fine-tuning models on clustered instructions. We find that the average density correlates with the test performance. Among the numbers, using 10 clusters achieves the highest performance.
>
> | The number of clusters                     | 5              | 10             | 15             | 20             |
> | ------------------------------------------ | -------------- | -------------- | -------------- | -------------- |
> | Average density of clusters                | 6.52           | 6.58           | 4.89           | 3.15           |
> | Average test performance over instructions | 79.62$\pm$1.62 | 80.96$\pm$0.85 | 78.29$\pm$0.34 | 77.20$\pm$0.77 |
>
> We note that determining the number of clusters is quick. Once we have estimated the task affinities, we can run the clustering algorithms with different clusters. The clustering step runs quickly, taking less than 1.5 seconds for the matrix of 100 tasks.
>
> We thank the reviewer for suggesting the non-parametric Bayesian processes. We have looked into the methods. From our understanding, such methods define probabilistic clustering structures as priors without the need to specify the number of clusters. We think it would be an interesting direction to design such processes in the context of clustering tasks in instruction fine-tuning. We will include a discussion of them in the related work.
>
> \>>> **Reducing the number of training samples in each task.**
>
> Thanks for the suggestion! Indeed, in the implementation of our approach, we downsample the training dataset of each task to speed up the training. We conduct an ablation study for the ratio of downsampling the training dataset. We use two pre-specified groups in our benchmark: sentiment classification and open-domain QA, which have the largest number of samples. We vary the ratio of downsampling between 0.01, 0.02, 0.05, 0.1, and 0.2. Then, we evaluate whether our clustering algorithm can identify the specified group.
>
> We observe that our algorithm accurately identifies the underlying group with a downsampling ratio of at least 0.05, below which the algorithm cannot identify the group. This suggests that the task affinity scores become more noisy when reducing more training examples. We will add this detail in the revised paper.
>
> \>>> **Clustering held-out tasks.**
>
> Thanks for the suggestion. As a preliminary study, we tested our approach on several held-out tasks. These include (i) structure-to-text generation tasks: E2E NLG, Web NLG, and Common Gen; and (ii) Sentence similarity classification tasks: STS-B, QQP, and WiC. We start with the task affinity matrix on the existing nineteen tasks in our task grouping benchmark, apply the adaptive sampling to estimate task affinity for the six new tasks, and then apply our approach to cluster the twenty-five tasks.
>
> We find that the structure-to-text generation tasks are grouped with existing summarization tasks, and sentence similarity classification tasks are grouped with coreference resolution tasks. This aligns with the categorization of the tasks: structure-to-text generation and summarization tasks are text generation tasks, and sentence similarity and coreference tasks are related to classifying whether two phrases/sentences are similar.

---

### Official Review · Reviewer_5Hpz · 2023-10-31

**Soundness:** 3 good
**Presentation:** 2 fair
**Contribution:** 2 fair
**Rating:** 3
**Confidence:** 3

**Summary:**

This paper proposes a novel approximate clustering algorithm, along with an evaluation benchmark, that aims to group tasks for language model instruction fine-tuning.

**Strengths:**

The new clustering algorithm achieves good fine-tuning results, which outperforms multi-instruction and prefix tuning by 3.3% on multiple tasks.

**Weaknesses:**

1) The paper utilizes SDP relaxation as an approximate clustering technique; however, it does not provide an analysis of the error bound that comes with using this method. To provide a better understanding of its applicability, it is essential to understand the worst-case scenario and where the SDP relaxation clustering algorithm fails.
2) The generalization of the SDP relaxation clustering algorithm is not analyzed in this paper, leaving questions about whether it is a general clustering solution that can approximate spectral clustering or if it only works for instruction fine-tuning.
3) While the paper proposes using an approximate solution, it does not provide clear explanation as to why the SDP relaxation clustering algorithm will outperform spectral clustering. Is it related to the clustering objective function definition issue?
4) It is not clear how to ensure that the output of the convex optimization can follow the ranking constraint. The paper could provide a more detailed exploration of this aspect of the algorithm.
5) The paper does not analyze the impact of different lambda values for different tasks. Is lambda the same in all experiments, or should it be tuned in each experiment?
6) The hyper-parameters, including k, m, s, and alpha, are not thoroughly studied. The paper could explore how to choose these values, such as whether they should be tuned for the particular task or selected according to some rules.
7) The paper proposes an adaptive estimation of task affinities; however, there is no analysis of how different sample sizes affect the final estimation accuracy. Examining how the estimation accuracy impacts the final results could provide deeper insights into the method's performance.
8) It would be helpful if the paper could provide the results of the spectral clustering approach and compare it with the multi-instruction and prefix tuning methods to give a more comprehensive assessment of the new clustering algorithm's performance.

**Questions:**

1) Please address the above weaknesses.

---

> ### Author Response · Authors · 2023-11-18
> **Response to Reviewer 5Hpz 1/2**
>
> We thank the reviewer for careful reading of our work and offering detailed comments. We respond to the comments of the reviewer below.
>
> \>>> **Error bound of SDP relaxation clustering algorithm.**
>
> Thanks for the great question. Empirically, we observe that on our task grouping benchmark, the SDP clustering algorithm correctly identifies the underlying groups in all 63 evaluation cases, obtaining the same results as the exhaustive search. Additionally, the SDP relaxation has been shown to recover optimal solutions under a planted cluster model in existing literature, such as Theorem 2.1 of Ames (Mathematical Programming 2014). We think that it would be interesting to study the limitations and worst case of the SDP relaxation clustering algorithm in task grouping settings.
>
> \>>> **Whether the SDP relaxation clustering algorithm is a general clustering solution or if it only works for instruction fine-tuning.**
>
> The clustering algorithm is derived on a non-negative weight matrix. Thus, it would be applicable to settings beyond instruction fine-tuning.
>
> \>>> **Why does the SDP relaxation clustering algorithm outperform spectral clustering.**
>
> We observe that the spectral clustering is sensitive when there is a large variation in the scale of cluster densities. This is because the eigenvectors used by spectral clustering are sensitive to the variations of scales among clusters. Such scenarios frequently happen in multitask learning when the loss scale of tasks varies dramatically.
>
> For example, we illustrate this with a synthetic matrix containing two high-density clusters (with mean of values $\alpha$) and a low-density cluster (with mean of values $\beta$). We find that when $\alpha$ is larger than $\beta$, the spectral clustering will not be able to separate the two high-density clusters. We also noticed that the eigenvector values within the high-density clusters remain constant. We illustrated this example in Figure 2 of the paper.
>
> \>>> **How to ensure that the output of the convex optimization can follow the ranking constraint.**
>
> To ensure the output satisfies the rank constraint, we conduct a rounding step on the SDP solution to generate $k$ clusters. Denote the SDP solution as $X$ and the number of tasks as $n$. We use a threshold $\lambda = \frac{c}{n}$, iterate through task $i=1$ to $n$, assign task $i$ and $j$ to a new cluster if $X_{i,j} \geq \lambda$, then start again from an unassigned task.
>
> \>>> **Setting the threshold lambda values for rounding the solutions.**
>
> We set the threshold $\lambda$ as $\frac{c}{n}$ where $c \geq 1$. Since the optimal value of $X_{i, j}$ should be $\frac{1}{|C|}$ where the $|C|$ is the size of one cluster, the value of $X_{i, j}$ should be larger than $\frac{1}{n}$ if $i$ and $j$ are in the same cluster. We tune the $c$ between 1, 1.5, 2, 2.5, and 3. We choose the $c$ such that it yields $k$ clusters. We find that setting $c=1$ mostly generates exact $k$ clusters.

---

> ### Author Response · Authors · 2023-11-18
> **Response to Reviewer 5Hpz 2/2**
>
> \>>> **Determining the hyper-parameters.**
>
> We discuss the process for determining each hyper-parameter below.
>
> - The number of clusters $k$: We choose $k$ that leads to the best average densities of clusters. We notice that the average density of clusters correlates with the resulting performance, thus making it a useful criterion.
> - The number of sampled subsets $m$: For the number of tasks $n$, we find that using $m = 5n$ samples is sufficient for the affinity scores to converge. In our algorithm, we estimate task affinity scores for additional $\frac{n}{s}$ tasks in each step. Thus, we sample $m = \frac{5n}{s}$ subsets.
> - The number of adaptive steps $s$: We observe that the results are stable using different numbers of steps. We ablate the number of steps between 5, 10, and 20 in multi-instruction fine-tuning experiments on RTE. We find that comparable performance is achieved when using different steps.
> - The size of subsets $\alpha$: We vary the subset size between 3, 5, 8, and 10. In the multi-instruction fine-tuning experiments, we notice that similar performance is achieved using different subset sizes. We also observe that using a larger $\alpha$ does not help since tasks with negative interference are more likely to be included in a larger subset.
>
> \>>> **Adding the results of spectral and Lloyd’s clustering.**
>
> Thanks for the suggestion. We have added the results of spectral and LLoyd’s algorithms for grouping instructions in multi-instruction fine-tuning. We set the number of groups the same as in our approach. Indeed, we find that our approach outperforms these two cluster algorithms by 5.3% on average over the five datasets. This aligns with our observation that spectral and LLoyd’s algorithm is sensitive to scale variations among clusters and can lead to wrong clusters.
>
> | Dataset                             | RTE                | WiC                | BoolQ              | E2E NLG            | Web NLG            |
> | ----------------------------------- | ------------------ | ------------------ | ------------------ | ------------------ | ------------------ |
> | Metric                              | Accuracy           | Accuracy           | Accuracy           | ROGUE-1            | ROGUE-1            |
> | Grouping by Spectral Clustering     | 73.18$\pm$0.45     | 65.09$\pm$0.67     | 75.71$\pm$0.55     | 71.91$\pm$0.26     | 81.27$\pm$0.94     |
> | Grouping by Lloyd's Clustering      | 73.58$\pm$1.03     | 64.61$\pm$0.64     | 75.61$\pm$0.30     | 71.26$\pm$0.42     | 80.41$\pm$0.49     |
> | Clustering by SDP relaxation (Ours) | **80.96$\pm$0.85** | **69.89$\pm$0.87** | **81.76$\pm$0.62** | **73.03$\pm$0.67** | **82.95$\pm$0.75** |
>
> We will incorporate the results into the revised paper.

---

> > ### Comment · Reviewer_5Hpz · 2023-11-23
> >
> > I appreciate the author's response as it addresses some of my original concerns. However, the detailed analysis of the approximate clustering algorithm still lacks. Its generalization requires more experiment results.

---

### Official Review · Reviewer_zYRS · 2023-10-31

**Soundness:** 2 fair
**Presentation:** 2 fair
**Contribution:** 2 fair
**Rating:** 3
**Confidence:** 5

**Summary:**

This paper starts from the well-known task grouping problem in Standley et al., 2020 and studies the following formulation: Given n tasks, find a partitioning of them into k groups so that each group of tasks can be best trained together (separately from the other groups).

**Strengths:**

+ The task grouping problem is indeed an important problem for language models

**Weaknesses:**

- The approach is missing a rigorous complexity analysis which for this topic seems very important
- The analysis seems incmplete. For xample "Ablation Studies," particularly in the subsection on "Instruction Selection," the authors compare the accuracy of the fine-tuned models resulting from their method and those obtained using other clustering algorithms. Considering providing a more comprehensive evaluation, it would be better if authors could also include a comparison of the time taken by each method.
- The clustering / aggregation approach and overall the prior works are poorly discussed and contrasted, especially recent mathematical approaches  that exploit advanced concepts like curvature and others.

**Questions:**

1) In the “Task Grouping Setup.” – page 3 and other parts of the paper, it would be useful if the authors could make it clearly whether the tasks are considered independent or there is some form of dependency and how it is captured beyond the task affinity matrix.
2) I have read many vague statements like “The computational cost of these techniques can still be quite high for the scale of instruction finetuning…” but no precise complexity analysis results. Can the authors provide at least in some cases concrete numbers?
3) The authors recognize that the clustering is a well-studied problem, they mention approaches based on SDP and Linear Programming relaxations, but it seems they have missed recent efforts on optimal transport theory like the Ollivier-Ricci curvature in “Ollivier-Ricci Curvature-Based Method to Community Detection in Complex Networks” (Scientific Reports 2019) or “Inferring functional communities from partially observed biological networks exploiting geometric topology and side information” (2022) and others which bear some analytical similitudes to LP relaxations. It would be fair to mention these theoretical works in clustering for weighted graphs as they are very much competitive or potential approaches for the problem studied here.
4) The fact that the “affinity matrix, in our case, can easily violate the triangle inequality” made me wonder if these Ollivier-Ricci curvature approaches that deal with advanced geometric concepts could be useful to this problem.
5) This is a minor issue, SDP is used in section 2 but only defined later.
6) In section 4.2, the authors should give the full name of the “MTL” to make readers better understand the meaning of “MTL performance”.
7) In Figure 2, the authors should include subscripts or labels to distinguish the results obtained from the three different methods.
8) The authors mention their use of high-order task affinity and adaptive sampling as methods to expedite the calculation of task affinities. In comparison to the direct training of n^2 models, it would be beneficial if the authors could quantify the time savings achieved through their proposed method. Clarification on this matter would greatly enhance the comprehensibility of the paper.
9) In Section 5.4, "Ablation Studies," particularly in the subsection on "Instruction Selection," the authors compare the accuracy of the fine-tuned models resulting from their method and those obtained using other clustering algorithms. Considering providing a more comprehensive evaluation, it would be better if authors could also include a comparison of the time taken by each method.

**Details Of Ethics Concerns:**

Not applicable.

---

> ### Author Response · Authors · 2023-11-18
> **Response to Reviewer zYRS**
>
> Thanks for reading through our paper and providing constructive feedback. We respond to the reviewer’s comments below.
>
> \>>> **Complexity analysis of task grouping approaches.**
>
> Thanks for the comment! We discuss the complexity of task grouping methods as follows. Let there be $n$ tasks, and we aim to partition the $n$ tasks into $k$ subsets.
>
> - The naive approach is to enumerate all possible combinations of tasks. This amounts to training and evaluating a model on $O(2^k)$ task combinations.
> - Prior works such as Standley et al. (ICML'20) and Fifty et al. (NeurIPS'21) have proposed to estimate pairwise task affinity scores and then apply a branch-and-bound-like search algorithm to find the best task combinations. However, the worst case of branch-and-bound search is still exponential.
> - Our work develops the clustering problem based on a semidefinite programming relaxation. Such semidefinite programs can be solved by interior methods with polynomial worst-case complexity (c.f. Section 5.7 of Lieven and Boyd (SIAM review'96)).
>
> In practice, we observe that using existing optimization tools to solve the SDP with $n=100$ tasks less than 1.5 seconds, while the branch-and-bound algorithm takes more than 24 hours for more than 20 tasks.
>
> \>>> **Discussion of two related works [1] [2]**
>
> Thanks for suggesting the reference! We have looked into the suggested papers and are happy to include them in the discussion of related works. From our understanding, these two works propose to compute Ollivier-Ricci Curvature for the edges of a graph and remove edges with negative curvature to reveal cluster structures. We think that it would be an interesting research question to evaluate and interpret such curvature given the transfer relationships between tasks in the context of task grouping.
>
> [4] Sia, Jayson, Edmond Jonckheere, and Paul Bogdan. "Ollivier-Ricci curvature-based method to community detection in complex networks." Scientific Reports (2019)
>
> [5] Sia, Jayson, et al. "Inferring functional communities from partially observed biological networks exploiting geometric topology and side information." Scientific Reports (2022)
>
> \>>> **Runtime of using high-order task affinity and adaptive sampling.**
>
> Thanks for the comment. We discuss the runtime of using high-order task affinity and adaptive sampling. Take multi-instruction fine-tuning as an example. There are 100 instructions regarded as $n=100$ tasks.
>
> - Directly estimating pairwise task affinities requires training $n(n-1)/2 = 4950$ models.
> - Estimating higher-order task affinities samples $m$ random subsets. We find that using the results of $m = 8n$ is sufficient for the affinity scores to converge. Thus, it requires training $800$ models.
> - Furthermore, using adaptive sampling divides the sampling into $s$ steps. Each step only considers $n/s$ new tasks and avoids sampling subsets across existing clusters. We find that in each step, it takes $m = 5\frac{n}{s}$ samples for the affinity scores to converge on average. Thus, in total, it requires training $500$ models. Taken together, using both methods leads to around $10\times$ less runtime compared to directly training $O(n^2)$ models.
>
> **Response to other comments.**
>
> \>>> **Runtime taken by other clustering algorithms.**
>
> We note that once we have estimated the task affinities, clustering on the task affinity matrix runs within a few seconds. For the affinity matrix of 100 tasks, our clustering algorithm with SDP relaxation takes 1.5 seconds, while spectral clustering and Lloyd’s clustering take 1.2 and 0.9 seconds, respectively.
>
> \>>> **Whether the tasks are considered independent or there is some form of dependency.**
>
> We consider that the tasks are independent of each other.
>
> \>>> **SDP is used in section 2 but only defined later; The authors should give the full name of the MTL; In Figure 2, the authors should include subscripts or labels to distinguish the results obtained from the three different methods.**
>
> We thank the reviewers for catching these typos. We have revised them accordingly in our revised paper.